# Reversible Dimerization of Human Serum Albumin

**DOI:** 10.3390/molecules26010108

**Published:** 2020-12-29

**Authors:** Alexey Chubarov, Anna Spitsyna, Olesya Krumkacheva, Dmitry Mitin, Daniil Suvorov, Victor Tormyshev, Matvey Fedin, Michael K. Bowman, Elena Bagryanskaya

**Affiliations:** 1Institute of Chemical Biology and Fundamental Medicine SB RAS, 630090 Novosibirsk, Russia; d.mitin@g.nsu.ru (D.M.); syvorov.daniil@gmail.com (D.S.); 2Novosibirsk State University, 630090 Novosibirsk, Russia; a.spitsyna@alumni.nsu.ru (A.S.); mfedin@tomo.nsc.ru (M.F.); 3N.N. Vorozhtsov Institute of Organic Chemistry SB RAS, 630090 Novosibirsk, Russia; 4International Tomography Center SB RAS, 630090 Novosibirsk, Russia; torm@nioch.nsc.ru; 5Department of Chemistry & Biochemistry, University of Alabama, Tuscaloosa, AL 35487, USA

**Keywords:** human serum albumin, pulse dipole EPR, aggregation

## Abstract

Pulsed Dipolar Spectroscopy (PDS) methods of Electron Paramagnetic Resonance (EPR) were used to detect and characterize reversible non-covalent dimers of Human Serum Albumin (HSA), the most abundant protein in human plasma. The spin labels, MTSL and OX063, were attached to Cys-34 and these chemical modifications of Cys-34 did affect the dimerization of HSA, indicating that other post-translational modifications can modulate dimer formation. At physiologically relevant concentrations, HSA does form weak, non-covalent dimers with a well-defined structure. Dimer formation is readily reversible into monomers. Dimerization is very relevant to the role of HSA in the transport, binding, and other physiological processes.

## 1. Introduction

Human serum albumin (HSA) is renown as the most abundant protein in plasma with multiple roles [1,2,3]: roughly 120–160 g of HSA is found at a concentration of roughly 45 mg/mL (or 0.7 mM) in the vasculature. However, an even greater amount of HSA, roughly 240 g, is found outside the bloodstream in the extravascular spaces of tissues, although at a lower concentration of about 20 mg/mL (0.3 mM) [2,3]. Yet, even when the volume of the tissue is small so that the HSA mass is small, it still can have important impacts; for example, in the cerebrospinal fluid HSA comprises 35–80% of the total protein [4,5]. HSA is quite soluble, but frequent encounters between HSA molecules provide an opportunity to form well-defined aggregates: dimers, trimers, or even larger structures. Many different proteins form aggregates that are a major factor in their biological function. For HSA, covalent dimers and larger aggregates, crosslinked by stable chemical bonds, are well known; but transient aggregation of HSA, at its high in vivo concentrations, is not.

While developing a new spin-label reagent [6], we noticed that some chemical reactions proceed differently at high HSA concentrations. That observation now leads us to consider whether the formation of HSA aggregates is involved. This question is significant far beyond the spin labelling field. HSA has several distinct physiological roles in several parts of the body where it has different concentrations. Any reversible formation of aggregates would have a considerable impact on each physiological process in which HSA participates. For convenience, we will refer to whatever small soluble, non-covalent aggregates form under physiologically relevant conditions as ‘dimer’ and address their composition and structure later.

One major role of HSA consists of the transport of many classes of small molecules. HSA has multiple sites that selectively bind their own specific range of small molecules or ligands and play an important role in transport, delivery, and elimination of those ligands. Among the ligands that bind at one or more sites are fatty acids; small peptides and proteins; hormones; metals; and drugs, such as warfarin and ibuprofen. This binding has also driven the pharmaceutical use of HSA as a drug carrier. An important feature of ligand binding is that the binding of a ligand at one site often affects the conformation of the HSA and alters the affinity of its other sites for their own ligands, an effect known as allostery, which has important consequences for pharmacokinetics, pharmacodynamics, and drug interactions [2,7,8].

HSA also has a vital role in the redox status of tissues. The cysteine amino acid at position 34 (Cys-34) is particularly noteworthy. In healthy individuals, up to 90% of the Cys-34 has a free thiol (-SH) group that provides a major reservoir of antioxidant activity against reactive oxygen species (ROS) and reactive nitrogen species (RNS) [1,2]. Oxidative stress modifies the Cys-34 thiol group into a disulfide bond with another HSA or small molecule; or converts it into a sulfenic, sulfinic, or sulfonic acid group. HSA contains ~80% of the free thiols in blood so that these post-translational modifications at Cys-34 can be used as a biomarker in several diseases involving ROS or RNS where Cys-34 thiol can drop to less than 20% [2,3,9,10,11]. The other 34 cysteines of HSA make up the 17 disulfide bridges that stabilize the protein structure and in general are not reactive.

Amino acids at several sites are activated in HSA, giving rise to a weak esterase or thioesterase enzymatic activity. These weak activities have physiological importance because of the large amount of HSA in the body [2,12]. Some reactions modify HSA itself and can serve as biomarkers or alter other properties of the protein. Yet another important role is the regulation of oncotic pressure of the blood and extravascular fluids. The large concentration and the charge of HSA help to maintain the balance in the volumes of fluids in the body, an activity that is very sensitive to dimers.

HSA, as produced, consists of 68% α-helices with little or no β-sheets with 3 domains. It has a total of 585 amino acids but no attached sugars or carbohydrates. During its long four- to five-week lifetime, HSA undergoes glycation, acetylation, esterification, oxidation, and other post-translational modifications that alter its properties [7,8,13]. Native HSA taken from blood is mostly monomeric, but also contains covalent dimers, trimers, and higher order oligomers. Some of the covalent dimers have a disulfide bond linking their Cys-34 [9]. However, the formation of non-covalent oligomers has been reported at or below physiological concentrations in response to extreme conditions of, e.g., pH, hydrodynamics, or temperature [14,15,16]. Such oligomerization of HSA, and the very similar analog—bovine serum albumin (BSA), typically leads to very large, insoluble oligomers with a substantial amount of β-sheet seemingly unrelated to the small reversible aggregates considered here under more physiologically relevant conditions.

Here we detect and begin to characterize reversible non-covalent HSA aggregates or “dimers” using the Pulsed Dipolar EPR Spectroscopy (PDS). The PDS technique PELDOR (also known as DEER) [17,18] has already been applied to HSA previously with great success to study its binding of fatty acids and its structure under extreme conditions [19,20,21,22]. We examine whether HSA in solution, under physiologically relevant conditions, forms significant amounts of non-covalent dimers; characterize their structure; and probe whether the monomer/dimer equilibrium or kinetics is affected by post-translational modification. PDS is analogous, in some respects, to Förster Resonance Energy Transfer (FRET). Both use dipolar interactions between transition dipoles to probe the distance between spectroscopic labels; both produce a response only when there is an interaction between two labels. FRET, however, uses electronic transitions of optical labels, while PDS uses EPR transitions of spin labels. An important difference is that in PDS, the relative orientation of the transition dipoles is controlled by the magnetic field, making it possible to quantitatively determine the distribution of distances between the labels (up to roughly a dozen nanometers) rather than just an average effective distance. The distance distribution immediately reveals the relative amounts of different conformers in a heterogeneous mixture of post-translationally modified HSA.

## 2. Results

### 2.1. Synthesis of Spin Labeled HSA and Detection of Dimers

Highly purified, native HSA free of bound fatty acids was fractionated by size exclusion chromatography adapted from Janatova et al. [23]. The ‘monomer’ fraction, composed of proteins with the ~ 66 kDa MW of HSA, was retained and will be referred to as m-HSA. Each portion of eluate was analyzed by SDS-PAGE (Appendix A). The HSA contains a broad spectrum of native post-translational modifications, including oxidation and disulfide bonds at Cys-34 which would interfere with subsequent spin-labelling. Therefore, the pooled monomer fraction was treated with dithiothreitol (DTT) to convert the modified Cys-34 into free thiol which is readily spin labelled, Appendix A [6,24].

HSA was labelled with two different spin labels, Figure 1: the classic MTSL having a nitroxide free radical; and the next-generation OX063-**17** label [6] having an OX063 trityl free radical. The labelled monomers are identified as m-HSA-XXX where XXX indicates the label—NIT, meaning nitroxide label, or OX063. Preparation and characterization are presented in Supporting Information for all samples. Both spin labels selectively labelled the Cys-34 thiol of 78 to 97% of the monomers. The OX063-**17** label has excellent water solubility which avoids the tendency of earlier trityl-based labels to form clusters of hydrophobic trityl groups [6]. The MTSL is also well behaved in polar aqueous media.

If the spin-labelled monomers do not form dimers in solution, then the distance from each spin label to the nearest one has a very broad distribution—producing a very smooth, flat, and featureless DEER signal. However, the DEER signals of both m-HSA-NIT and m-HSA-OX063 show a peak at short times (> 150 ns) approaching a constant level, sometimes after a few oscillations, at long times, Figure 2, left-hand column. The strong peaks, and especially the oscillations, show unequivocally that some HSA monomers form dimers (or larger aggregates) and not a homogeneous solution.

These DEER signals can be converted from a function of time into the distribution of distances between adjacent labels, Figure 2, see Supporting Information. Both spin labels yield a distance distribution having a clear peak near 2 nm. The m-HSA-OX063 has a sharp peak at 2.1 nm with a width of about 0.3 nm, indicating that the center carbons of the two trityl radicals are 2.1 ± 0.3 nm apart in the dimers. Each trityl group has a core radius of about 0.5 nm, so they cannot be in physical contact. Their positions are constrained by the protein surface and by the Cys-34 via the molecular linkers of the spin label. The dimer contains at least a very ordered pair of monomers to have such a narrow distance distribution. Noise in all the DEER signals does perturb the distance distributions; the shaded regions indicate the uncertainties. Beyond ~3 nm the features in the distributions with either label are comparable to the uncertainty.

The distance distribution for m-HSA-NIT has a much broader peak at ~1.9 nm. The nitroxide label has a much shorter linker compared to the OX063 label, yet the dominant peak near 2 nm shows a much broader distribution of distances between labels. This apparent contradiction occurs because the nitroxide group is much smaller than the trityl group in the OX063 label. It is so small that it can displace amino acid sidechains and insert itself into the protein with different linker conformations while the trityl label is excluded from the protein. Even if the Cys-34 attachment points are at fixed distance apart, the distance between the free radical ends of the labels, which is the quantity measured in DEER, can vary as much as four times the length of the linker, an effect seen in other studies using MTSL [17]. No additional intense peaks, indicative of well-ordered aggregates of three or more HSA, are seen at larger distances. Thus, the ‘dimers’ likely are a well-ordered dimer, or perhaps a poorly-ordered cluster of well-ordered dimers.

The DEER spectra and their distance distributions clearly show well-ordered dimers in both sets of monomer samples, so it becomes crucial to determine whether these dimers are rare transient encounters between proteins or a significant population in the samples. This can be resolved from the normalized amplitude change in the DEER signal, which is called the modulation depth. Only spins, therefore, spin labelled HSA molecules contribute to the DEER signal. Thus, modulation depth is proportional to the amount of spin in pairs relative to the total number of spins in the sample. For m-HSA-NIT, the DEER signal goes from 100% at time zero to 93% at long times, a normalized change of 7%, see Appendix A. This change is roughly the fraction of dimers in the sample scaled by the instrument response, where normalized change of the DEER signal for 100% pairs equals 23% for MTSL and 25% for OX063 labels. Therefore, roughly 30% (7%/23%) of m-HSA-MTSL form dimers containing two labels, which hereinafter will be referred to as doubly labelled dimers. Similarly, the 11% normalized change for m-HSA-OX063 corresponds to about 44% in doubly labeled dimers, Table 1. Dimers are a significant part of both samples, at sample concentrations that are in the physiological range.

### 2.2. Dilutions

The equilibrium between monomer and dimer was probed by 1:1 dilution of labelled samples by unlabeled HSA previously treated with DTT, so that total protein concentration was maintained. The samples were diluted, thoroughly mixed, and incubated one hour at 37 °C to ensure equilibration. Undiluted, similarly incubated samples served as controls. The diluted samples had no significant changes in the distance distributions compared to undiluted controls, Figure 3 and Figure 4. However, there was a significantly smaller DEER modulation depth, indicating that dilution with unlabeled m-HSA, while keeping total protein constant, decreased the fraction of labelled m-HSA in doubly labelled dimers. Such a decrease is expected if monomers are in a dynamic equilibrium with dimers and corroborates the DTT-free PAGE, Appendix A, and size exclusion chromatography assays showing an absence of covalently linked dimers and oligomers in the spin labelled samples. The dimers are in equilibrium with monomers so that dilution with unlabeled m-HSA converts many of the doubly labelled dimers into mixed dimers having one labelled and one unlabeled HSA, which consequently have no DEER signal.

The dilution of m-HSA-NIT caused a roughly 30% decrease in fraction of doubly labelled dimer, but only a 20% decrease for m-HSA-OX063, Table 1. These decreases are both less than the 50% expected statistically and suggest that there is an allosteric effect on dimer formation. The small nitroxide label can be expected to perturb the HSA conformation to a similar degree as the binding or covalent attachment of other small molecules; and to produce allosteric effects, as do many post-translational modifications and ligand bindings. That the decrease is less than 50% in doubly labelled dimer shows attractive interactions between two labelled HSAs that are stronger than those between a labelled and an unlabeled HSA. Similar reasoning indicates somewhat stronger interactions between m-HSA-OX063 than between m-HSA-NIT.

### 2.3. Ligand Binding

Myristic acid is one of the many fatty acids, drugs, and small molecule ligands that HSA transports in the body. Binding of many of these ligands affects the conformation of HSA and its binding affinity for other ligands. We therefore used myristic acid to determine whether dimer formation is affected by the binding and subsequent transport of small ligands. Myristic acid was added to the monomer samples and incubated one hour at 37 °C to ensure equilibration. A 1:1 ratio of myristic acid had no detectable effect on the DEER signals. However, when myristic acid concentration was increased to 25:1, the DEER modulation depth decreased, but the signal shape and the distance distribution remained unchanged, Figure 3B and Figure 4B.

There is no difference in the structure of the dimers in the presence of myristic acid, but the dimers decreased to about 0.6 of their initial number. The dimers are less stable and more prone to dissociate into monomers when myristic acid binds. However, the converse is also true and could be even more significant. Formation of a dimer promotes release of ligated myristic acid while dissociation into monomers promotes uptake.

These experiments show that ligand binding does affect dimerization. However, HSA has multiple ligand binding sites with different affinities for different types of ligands. It is not clear which site(s) are involved in these allosteric effects of myristic acid. Different sites and other ligands may well have different effects.

## 3. Discussion

### 3.1. Dimer Structure

Non-covalent dimers form in substantial numbers in solutions of spin-labelled HSA monomers at concentrations and pH that are physiologically relevant. In these dimers, the spin labels attached to the Cys-34 of each monomer are 1.9 nm apart for the MTSL label and 2.1 nm apart for the OX063 label. The distribution of distances was much sharper, ± 0.3 nm, with the OX063 label, partly because of a less noisy signal and partly because the position and conformations of the larger OX063 label are more tightly constrained when bound to the protein. The labels are positioned rather precisely relative to each other. Cys-34 sits in a hydrophobic crevice 0.95 to 1.0 nm below the surrounding surface of the HSA. The nitroxide group can find a niche within the protein near Cys-34, but the trityl group of OX063 is much larger and must be accommodated in the crevice. If the labels are precisely positioned relative to each other, the dimer containing them must be well-ordered with their Cys-34 relatively close and they must have a definite structure that retains much of secondary and tertiary structure of the monomers because CD and DLS studies show they remain largely intact, see Appendix A.

Two covalently linked dimers are known which hold the Cys-34 close to each other while retaining the overall structure of the monomers intact. The covalent dimers indicate how dimer formation can affect properties and behavior of the monomers. In one of these dimers, the Cys-34 are directly connected by a disulfide bond [2,3]; the other dimer is chemically produced by crosslinking the Cys-34 via 1,6 bis(maleimido)hexane (BHM) [25]. In these two covalent dimers, the Cys-34 are close to each other and reside near the monomer surface, so they provide useful benchmarks for a discussion of the non-covalent dimer found at high HSA concentrations.

HSA will form non-covalent dimers and other oligomers when subjected to high temperatures, pH extremes, and other major stresses [14,15,16,21,22,26,27,28]. Some are irreversible and nearly all have major decreases in α-helical content and large increases in β-sheet that indicate major structural changes. The dimers of HSA found here are quite different. Dimer formation is reversible, and dimers readily revert to monomers when diluted for CD, dynamic light scattering, PAGE and size exclusion chromatography assays. CD spectra show that the α-helical structure is maintained in the dimers, with no major increase in β-sheets, Appendix A, quite different from the behavior reported for HSA in extreme conditions. The dimer studied here seems to retain the structure of its individual monomers and does not seem to be like other reported non-covalent dimers and oligomers produced in very non-physiological conditions.

### 3.2. Dimer Properties

Detailed crystal structures have not been determined for the covalent dimers just mentioned. However, some of their physical properties have been studied and provide some indication of possible consequences of dimer formation. The disulfide dimer requires some molecular distortion of the monomers to pull each Cys-34 out of the crevice in which it sits and to bring them close enough to form a direct disulfide bond. In this dimer, some binding sites are unable to bind ligands, e.g., binding of tryptophan is blocked [25]. The entrance to the binding site may be covered in the dimer, or the structural distortions produced in the direct dimer may induce allosteric changes in binding or kinetics. Crosslinking with BHM (roughly 1.6 nm in length [29]) does not require major distortions and leaves binding constants of Sites I or II for warfarin and diazepam, respectively, ”comparable” in the crosslinked dimer to those of monomers [25].

The non-covalent dimer could be considered somewhat intermediate between these two covalent forms. A dimer can be formed by juxtaposition of the Cys-34 of two monomers, shown schematically in Figure 5. According to the distance between the two radicals in albumin molecules ~ 2 nm we can propose the interaction of Domain I, precisely subdomains IA and IB, of two HSA molecules (Cys-34 is in subdomain IA). The close structure of the dimer was proposed and published for bovine serum albumin [30]. However, BSA has two cysteines at residues 34 and 513 that can form intermolecular disulfide bonds connecting two BSA monomers. For BSA structure cysteine at residue 513 could form a molecular hinge between the BSA monomers and stabilize the BSA dimer [30]. The disulfide bridge in the BSA dimer defines a rotational axis that is visible as well in the normal mode displacement patters. Therefore, for the HSA we proposed a similar dimer structure. Such a dimer has no covalent bond that must be formed so no large, local distortions are needed to accommodate it. In that respect, it resembles the cross-linked dimer.

On the other hand, the non-covalent dimer is held together by interactions across a substantial interface between the monomers. The surface of HSA around the crevice containing Cys-34, and its pendant spin label, is covered by patches of positive and negative charges which could electrostatically stabilize the dimer. However, the opposite charges need to be aligned and in contact, which may require conformational adjustments having allosteric effects. The large area of contact can also seal the entrance to binding sites on that surface, preventing ligand release and/or uptake until the dimer dissociates. HSA sites having enzymatic activity will experience similar effects, modulating their activity as the monomer/dimer equilibrium shifts. Thus, the important binding, transport, and catalytic activities of HSA can be modulated to various degrees by the dynamic formation and dissociation of dimers.

The coupling between dimerization and function is clearest for the contribution of HSA to oncotic pressure. The large concentration of HSA in plasma and in the extravascular fluids gives it a significant role in balancing fluid distributions and in maintaining blood volume. The most relevant concentration is not the grams of HSA per liter, but the number of HSA particles per liter. Two monomers are twice as potent as a single dimer [25,31].

The surface of HSA at the interface has areas of positive and areas of negative charge that can help stabilize the dimer via electrostatic interactions and lock the two monomers into a well-ordered structure that is observed. Some of the charged groups in HSA are post-translational modification sites. Any such modifications at the interface between monomers that alter charge will modulate the electrostatic interactions and affect the stability of the dimer.

In fact, many post-translational modifications or ligand bindings produce allosteric effects by causing slight conformational changes in the HSA. Some will even propagate to the interface between monomers and affect stability of the dimer. This was seen in the dilution experiments, where the spin label can be considered a post-translational modification, albeit unnatural, and in experiments with myristic acid. The changes in amounts of dimers formed show that dimer stability depends to some degree on the chemical modifications and the ligands bound to each of the monomers in the dimer. It is important to remember that the HSA in these experiments was extracted from native HSA and contains a wide variety of post-translational modifications. There is a variety of different dimers and the monomers in a single dimer may have different sets of modifications. Consequently, each sample will have a complex equilibrium between many variants of HSA, which is not described by a unique equilibrium constant.

However, we can estimate an apparent dissociation constant K_D_ lying in the range 0.1–1.0 mM from the relative amounts of labelled monomers and doubly labelled dimers seen in the DEER measurements. This range coincides well with the range of physiological concentrations in different tissues [2,3]. So, whenever the HSA concentration changes, e.g., in moving from blood in the vasculature (at ~0.7 mM) to extravascular spaces of tissues (0.3 mM), it will cause a significant redistribution of HSA between dimers and monomers. In turn, the allosteric effects in ligand binding will trigger an accompanying uptake or release of ligands, including drugs, hormones, and small peptides.

### 3.3. Dimers In Vivo

We find that HSA forms dimers in large amounts at concentrations and pH that are physiologically relevant, and with a broad variety of natural post-translational modifications. However, the experimental solutions lack all other proteins and virtually all the small molecules found in bodily fluids. So, even though this work shows that HSA can form dimers in solution, the vital question of whether such dimers do exist in bodily fluids remains. Even though one may expect effects like protein crowding to encourage even more dimerization in plasma and extravascular fluids, experimental verification can be complicated and involved because formation of dimers of other proteins or heterodimers of HSA with another protein must be excluded. This can be difficult because HSA lacks unique properties, such as a strong visible absorption spectrum, that would distinguish it and its dimers from all other proteins. DEER measurements are not attractive for detecting in vivo dimerization because of the need to attach a spin label to nearly all the HSA for good signal amplitude. The labeling must be free of side reactions, highly specific, and must not perturb properties of the HSA. One possibility is to use a strong ligand binding site of HSA to bind a spectroscopic label that can be used with other methods to resolve the diffusion of monomers from that of dimers, e.g., by light scattering or NMR methods.

HSA is an important hub where many physiological processes, pathologies, diagnostics, and therapeutics intersect [2,3]. The HSA dimers and their interactions with the many allosteries of HSA, may provide the framework for understanding more fully the regulation of processes involving HSA, as well as, the detection and treatment of diseases.

In several diseases, Cys-34 is extensively modified and the amount of Cys-34 thiol plummets. These extensive modifications are attractive potential biomarkers to follow disease progression and therapy. However, these modifications can play a direct role in the disease if the modification of Cys-34 affected the monomer/dimer equilibrium and, in turn, the uptake and release of important ligands, or the regulation of oncotic pressure. In this work, the spin labels were attached to Cys-34 which made it impossible to measure the impact of natural modifications of Cys-34 on dimerization of HSA. Attachment of spin labels at other sites in HSA [32], combined with introduction of disease-related modifications at Cys-34 or other relevant sites would allow future DEER experiments to investigate this fascinating possibility of a more direct involvement of HSA in such diseases.

HSA dimers exit the blood vessels into the extravascular fluids as a normal part of their circulation. The extravascular fluids contain much lower levels of HSA, so the equilibrium shifts toward more monomers and fewer dimers. As the dimers dissociate, their binding affinities for ligands, their enzymatic activities, their oncotic pressure contributions, etc. also change. Understanding and modeling of any physiological processes in which HSA takes part requires the consideration of HSA dimer formation.

HSA dimers may offer new ways to alter or control some functions of HSA. The dilution and myristic acid binding studies indicate that the monomer-dimer equilibrium is sensitive to the binding of ligands to HSA or to post-translational or chemical modifications at Cys-34 and perhaps other sites as well. This provides a route for influencing the extent of dimer formation and all the functions of HSA that are modulated by dimer formation. For example, a drug, molecular signal or chemical modification that made dimer formation more likely would consequently enhance ligand binding and transport that was favored in dimers, providing possibilities for the body to regulate functions of HSA, for intervention in disease progression, or for targeted drug delivery.

## 4. Conclusions

Non-covalent dimers were detected, and their structure characterized at physiological concentrations using DEER spectroscopy. Monomeric HSA was separated by SDS-PAGE and then spin labeled with either MTSL or OX063. Dilution experiments with unlabeled HSA indicated non-covalent HSA dimers in equilibrium with monomeric HSA. The binding of myristic acid as a ligand to HSA dimers affected monomer-dimer equilibrium, as did the chemical modification of Cys-34. When diluted for CD, dynamic light scattering, PAGE and size exclusion chromatography assays, dimers readily dissociated into monomers.

CD assays find that the α-helical structure is preserved in the dimers, with no major increase in β-sheets. Distance distributions indicated well-ordered dimer structures with the mean distance between spin labels of 1.9 nm for MTSL and 2.1 nm for OX063. We attributed this difference to MTSL label having smaller size and a shorter linker compared to OX063 label. Dimer formation seems to have allosteric effects on HSA and could influence the many physiological function of HSA.

## Figures and Tables

**Figure 1 molecules-26-00108-f001:**
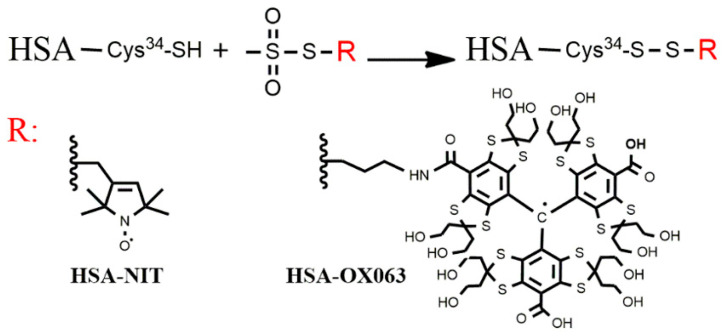
Spin labelling of HSA. Reaction of the methanethiosulfonate group of the spin label reagent with the thiol group of Cys-34 to generate spin-labelled HSA, R: nitroxide or OX063 labels.

**Figure 2 molecules-26-00108-f002:**
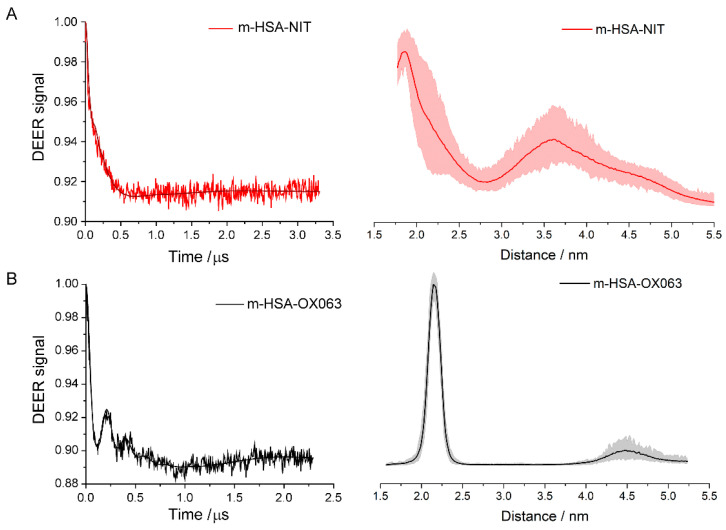
Q-band DEER results for (**A**) m-HSA-NIT sample; (**B**) m-HSA-OX063 sample. Background-corrected DEER time traces are on the left, distance distributions on the right. Shaded areas show uncertainty of distance.

**Figure 3 molecules-26-00108-f003:**
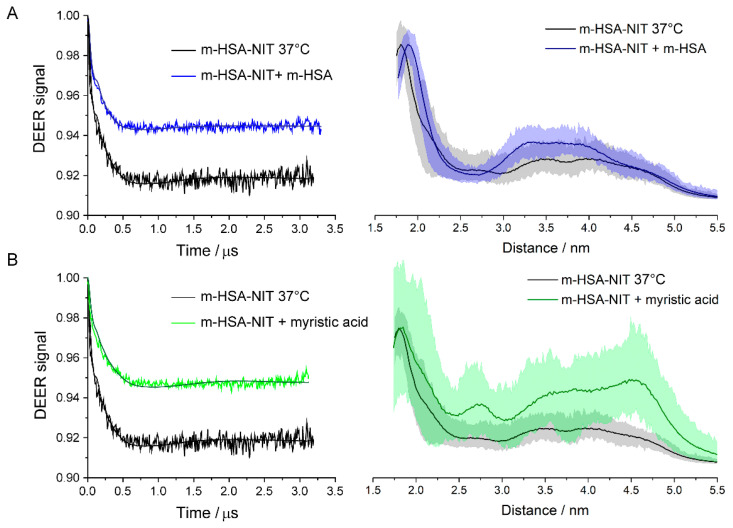
Q-band DEER results for dilution and myristic acid addition to m-HSA-NIT compared with sham controls incubated for 1 h at 37 °C. (**A**) control compared with 1:1 dilution with unlabeled m-HSA; (**B**) control compared with 25:1 excess of myristic acid. DEER time traces are shown on the left, distance distributions on the right. Shaded areas show uncertainty of distance distributions.

**Figure 4 molecules-26-00108-f004:**
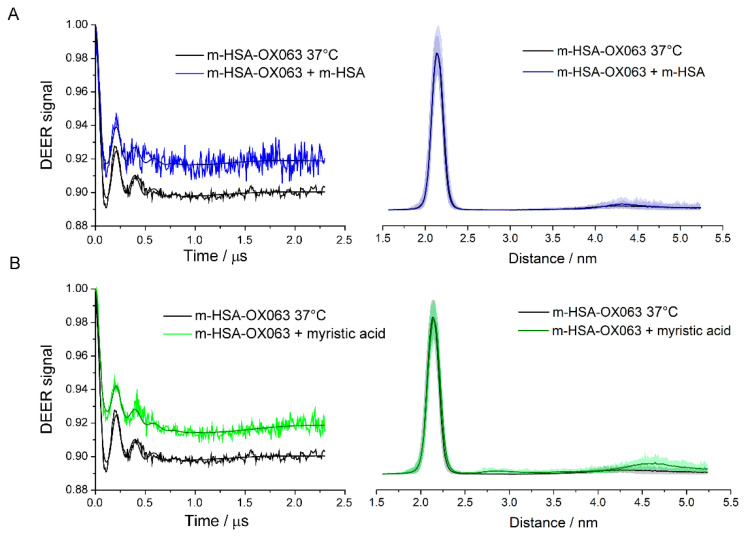
Q-band DEER results for dilution and myristic acid addition to m-HSA-OX063 compared with sham controls incubated for 1 h at 37 °C. (**A**) control compared with 1:1 dilution with unlabeled m-HSA; (**B**) control compared with 25:1 excess of myristic acid. DEER time traces are shown on the left, distance distributions on the right. Shaded areas show uncertainty of distance distributions.

**Figure 5 molecules-26-00108-f005:**
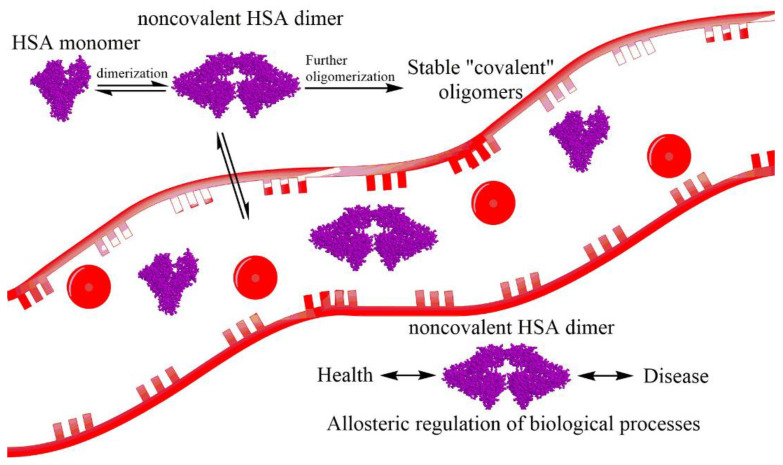
Structure of Human Serum Albumin (HSA) and schematic representation of self-oligomerization of albumin with further formation of stable dimers. Noncovalent dimer can affect the redistribution of albumin along with the organism. It, perhaps, has an allosteric effect and regulation on several biological processes.

**Table 1 molecules-26-00108-t001:** Percentage of spin labels per HSA and content of dimer of two spin labelled HSA as % of total labelled HSA content.

Sample	Spin Labels per HSA Molecules %	Spin Labeled HSA Concentration 10^−4^ M	Total HSA Concentration 10^−4^ M	Dimer of Two Spin Labelled Monomers %
m-HSA-NIT	97	6.79	7.0	30
m-HSA-NIT: m-HSA = 1:1	45	3.15	7.0	20
m-HSA-OX063	79	11.06	14.0	44
m-HSA-OX063: m-HSA = 1:1	39	3.9	10.0	32

## Data Availability

Not available.

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
