# Peer review of "Reversible Dimerization of Human Serum Albumin"

_molecules, 2020, doi:10.3390/molecules26010108_

Round 1
Reviewer 1 Report
In this work authors present measurements about the reversible dimerization of HSA applying different methods. The work is interesting and surely will earn wide interest due to the importance in several part of biology. These properties justify publication in Molecules. The references are up to date and comprehensive. I can suggest publication after considering the following minor remarks:
Association of large biological molecules are known to support by the hydrophobic effect regarding the increased freedom of water molecules during the interaction. How this process can affect the interaction authors examined.
The thermodynamic parameters would be most relevant according to the association. I see that the very weak interaction cannot offer enough information to calculate the enthalpy and entropy term, but this case the conclusion also affected by several parameter which surely changes when the molecule travels from a site to another. What is the opinion of the authors, how this property can affect their conclusion?
Author Response
In the present study, we propose the structure of the noncovalent albumin dimer. The information is presented in the graphical abstract (see Picture) of the paper and Fig. 5 in the main paper file.
According to the distance between the t wo radicals in albumin molecules ~ 2 nm we can propose the interaction of Domain I, precisely subdomains IA and IB, of two HSA molecules (Cys-34 is in subdomain IA). The close structure of the dimer was proposed and published for bovine serum albumin (see Picture) [1]. However, BSA has two cysteines at residues 34 and 513 in BSA that can form intermolecular disulfide bonds connecting two BSA monomers. And for BSA structure cysteine at residue 513 could form a molecular hinge between the BSA monomers and stabilize the BSA dimer [1]. The disulfide bridge in the BSA dimer defines a rotational axis that is visible as well in the normal mode displacement patters. Therefore, for the HSA we proposed a similar dimer structure.
[1] Ameseder, F.; Biehl, R.; Holderer, O.; Richter, D.; Stadler, A.M. Localised contacts lead to nanosecond hinge motions in dimeric bovine serum albumin. Phys. Chem. Chem. Phys. 2019, 21, 18477–18485, doi:10.1039/c9cp01847f.
We add this explanation and reference 1 into manuscript.

Reviewer 2 Report
By using PDS-EPR, the authors showed that HSA is in equilibrium between monomer and dimer, the dimer contains the same conformation of the monomer, and the ligand binding can modulate the equilibrium. Because HSA exists at high concentration in physiological condition (> 0.7 mM in blood), it is likely that HSA can form higher order oligomers. In this work, the presence of non-covalent dimer of HSA was proven experimentally.
One minor suggestion to the authors is the addition of discussion about the orientation and ligand binding of monomers in dimeric HSA. The reviewer think that if the distance between the spin labels in dimer is about 2 nm, then the associated pattern or organization of monomers in dimer can be predicted with known monomeric structure of HSA. Even though the authors presented the schematic structure of dimer in Fig 5, with more detailed organization of monomers in dimer, it will be better to discuss the ligand binding of noncovalent dimer in comparison with those of covalent or crosslinked dimer.
Author Response
The reviewer is right that hydrophobic properties of large biological molecules can support the aggregation process. However, DEER only provides information about distances between spin labels on nanometric scale, and it is impossible to make conclusions about HSA aggregation process based on DEER data only. Moreover, DEER experiments are conducted in frozen solution at cryogenic temperatures, thus, it is not possible to obtain any thermodynamic parameters.